# Amine Infused Fly Ash Grafted Acrylic Acid/Acrylamide Hydrogel for Carbon Dioxide (CO_2_) Adsorption and Its Kinetic Analysis

**DOI:** 10.3390/gels9030229

**Published:** 2023-03-15

**Authors:** Siti Musliha Mat Ghani, Nurul Ekmi Rabat, Abdul Rahman Abdul Rahim, Khairiraihanna Johari, Ahmer Ali Siyal, Rowin Kumeresen

**Affiliations:** Chemical Engineering Department, Universiti Teknologi Petronas, Seri Iskandar 32610, Perak, Malaysia

**Keywords:** amine infused hydrogel, carbon dioxide, adsorbent, fly ash, kinetic study, monoethanolamine, acrylamide, acrylic acid, greenhouse gas

## Abstract

In most carbon dioxide (CO_2_) capture processes, chemical absorption using an amine solvent is widely used technology; however, the solvent is prone to solvent degradation and solvent loss which leads to the formation of corrosion. This paper investigates the adsorption performance of amine-infused hydrogels (AIFHs) to increase carbon dioxide (CO_2_) capture by leveraging the potency of amine absorption and adsorption properties of class F fly ash (FA). The solution polymerization method was used to synthesize the FA-grafted acrylic acid/acrylamide hydrogel (FA-AAc/AAm), which was then immersed in monoethanolamine (MEA) to form amine infused hydrogels (AIHs). The prepared FA-AAc/AAm showed dense matrices morphology with no obvious pore at the dry state but capable of capturing up to 0.71 mol/g CO_2_ at 0.5 wt% FA content, 2 bar pressure, 30 °C reaction temperature, 60 L/min flow rate, and 30 wt% MEA contents. Cumulative adsorption capacity was calculated and Pseudo-first order kinetic model was used to investigate the CO_2_ adsorption kinetic at different parameters. Remarkably, this FA-AAc/AAm hydrogel is also capable of absorbing liquid activator that was 1000% more than its original weight. FA-AAc/AAm can be used as an alternative AIHs that employ FA waste to capture CO_2_ and minimize the GHG impact on the environment.

## 1. Introduction

One of the world’s key focus is to minimize greenhouse gas (GHG) emissions impact on climate resilience and environmental sustainability. GHG such as CO_2_ accumulate in the atmosphere and cause climate change by trapping heat. The main GHG emission contributors are the transport, manufacturing industries, industrial processes, waste, agriculture sector and energy industries. As of July 2022, statistics from Global Energy Monitor shows there are more than 2215 power plants that actively generate electricity from the burning of coal [1]. A conventional post-combustion capture technology (PCC) used in plants normally utilize amine as the absorbent solvent that binds with CO_2_. Amine as a solvent is generally used for the removal of CO_2_ as it has a faster rate of reaction with CO_2_. However, the main problem related to the use of amine as a solvent for the CO_2_ absorption is its high cost [2]. The high cost is associated with its regeneration cost which results in only 50% of the solvent at the end. Another problem with amine solvent is it affects operating problems such as solvent degradation, solvent loss and corrosion to the equipment [3].

Adsorption is an appealing method because it requires little energy to regenerate and is easy to process. However, it has limitations when used in high moisture environments and at low pressure conditions, which can affect its capacity and selectivity [4,5]. To improve its performance, amine-functionalized adsorbents were introduced. Amine-functionalized adsorbents are prepared by attaching amines to a porous support, either through in situ formed polymeric amines or impregnation route. Impregnated route adsorbent is created by physically adsorbing amines onto the support, while amine-grafted/covalently bonded systems are made through chemical reaction during synthesis or post-synthesis modification [4].

For instance, Chen et al. (2017) employed a post-functionalization technique to introduce amines into silica and observed that the CO_2_ adsorption capacity remained stable even after 10 cycles of adsorption [6]. Liu et al. (2019) developed a thermosensitive solid amine CO_2_ adsorbent by grafted polyethyleneimine (PEI) on the poly(*N*-isopropylacrylamide) via inverse suspension polymerization and it recorded 3.15 mmol/g of CO_2_ adsorption at 10 °C under wet condition [5]. Similarly, Choi et al. (2022) employed the in-situ method to bond hyperbranched poly(amidoamine)s on the ethylenediamine via oil-water-oil suspension polymerization and successfully formed microporous hydrogel particles that exhibited a CO_2_ adsorption capacity of 104.4 mg/g from flue gas in its dry state. The average pore size of the particles also varies from 2.9 µm to 1.3 µm depending on the agitation speed [7]. In another study, Kim et al. (2021) physically impregnated MEA into the dried hydrogel, resulting in an amine infused hydrogel (AIHs) with increased CO_2_ selectivity in the methane/CO_2_ mixture gas system compared to bulk MEA solution [8]. White et al. (2021) also utilized the same method to infused the amine into the hydrogel and investigated the use of water and ethylene glycol (EG) as the amine solvent in order to increase the uptake kinetic and minimize solvent degradation after adsorbent recycling. The study found that a combination of EG-diethanolamine resulted in a CO_2_ capture capacity of 0.75 mol CO_2_ per mol amine and that the adsorption capacity of the AIHs was highly dependent on the type of hydrogel used, with poly(acrylamide) found to be more effective in capturing CO_2_ compared to poly(*N*-2-hydroxythyleneacrylamide) and sodium polyacrylate [9].

Hydrogels, as a solid support, possess unique characteristics such as the ability to retain high amounts of liquid while maintaining their structural integrity through chemical or physical crosslinking of the polymer chains [10]. In order to exploit its potential as a CO_2_ adsorbent, the properties of hydrogels can be manipulated by infusing amine solvents into the hydrogel matrix. This AIHs can be prepared through the impregnation route, which is a simple and efficient process that reduces both operation time and capital costs [11]. The AIHs act as microcontainers when CO_2_ molecules diffuse through the surface of the hydrogel and react with the amine solution in-situ. This interaction with CO_2_ results in the formation of carbamates or bicarbonates. The AIHs exhibit a combination of physical adsorption inherited from the parent adsorbent and chemical adsorption from the loaded amine groups, resulting in a much higher CO_2_ adsorption capacity when compared to separate microcontainers and bulk amines [12]. Additionally, the use of AIHs technology eliminates the need for extra packing materials, as the hydrogel itself serves as the “amine storage and packing” [11].

The utilization of solid support mechanisms in the development of gas adsorbents to minimize the drawback of amine solvent has been a topic of ongoing research for an extended period of time [13,14,15,16,17,18]. However, the use of power plant generated waste, specifically fly ash (FA) as an additive in polymer adsorbents to enhance gas adsorption properties, has not been widely explored in the research literature. Previous FA adsorbent studies primarily focused on the surface treatment of the FA itself and it was proven that chemically treated FA did improve the surface area, morphology, porosity, and adsorbent affinity of fly ash [19]. The enhancement of CO_2_ adsorption by FA can been done via various methods such as alkaline activator treatment [20,21], conversion of FA into zeolite [22,23] and impregnation of raw FA using amine such as MEA [24]. For instance, this solid waste was transformed into mesoporous adsorbent called silica-alumina composite aerogel with 400 m^2^/g of active surface area and 1.9 cm^3^/g of specific pores volume via alkaline fusion and acid leached method of FA. It able to adsorb 2.02 mmol/g of CO_2_ [25] compared to the pristine FA which only can capture CO_2_ up to 870.1 µmol/g FA [20]. FA is the coal power plant byproduct that trapped at the end of the process using precipitator or baghouse [26]. It contains high carbon content, silica dioxide (SiO_2_) and aluminum oxide (Al_2_O_3_) that can facilitate the CO_2_ adsorption [20]. Within the last 20 years, an average quantity of 64.8 Mts FA was produced every year by the coal power plant in United State (US) and the highest utilization of FA was only at 60% which is in 2017 [27].

In this work, newly explored FA grafted AIHs was fabricated via direct impregnation of MEA in FA grafted AAc/AAm hydrogel. The impact of FA content, type of hydrogel activator, and operational temperature on CO_2_ capture was systematically evaluated in order to maximize the CO_2_ adsorption capacity. To gain a deeper understanding of the CO_2_ adsorption mechanism of MEA-infused FA grafted AAc/AAm hydrogels, adsorption kinetic model analysis was performed using five different kinetic models. The results of this work suggest that the utilization of FA to enhance the adsorption properties of amine-infused hydrogels is a promising area for future investigation.

## 2. Results and Discussion

### 2.1. Characterization of Materials

#### 2.1.1. Fly Ash Characterization

Table 1 presents the chemical composition of FA. The composition of FA varies depending on the source of the FA. In this study, the coal FA that obtained from local power station recorded the chemical composition of silicon oxide (SiO_2_), aluminum oxide (Al_2_O_3_) and iron oxide (Fe_2_O_3_) were 78.58 wt%, which indicated that the currently used FA was classified as a Class F FA as described in ASTM C618. The Class F FA contains more than 70% of Al_2_O_3_, SiO_2_ and less than 10% of CaO. While some analysis reported that the composition of SiO_2_ and Al_2_O_3_ can be up to 55% and 35% respectively [28,29]. The properties of the FA are highly dependent on its composition especially silicon (Si). High amount of silicon is able to contribute to high mechanical strength and adsorption capacity of the synthesized material [30,31]. On the other hand, the FA will be categorized as Class C if the total of Al_2_O_3_, SiO_2_ and ferrite is less or equal to 50% of the total FA composition and the composition of CaO is more than 10% [29].

Figure 1 shows the particle size distribution of raw FA. The particle size distribution of the FA was measured by the laser diffraction particle size analyzer. The particle size of FA ranged from 0.55 to 440 µm with median, D_50_ of 15.37 µm, D_90_ 90% of the particles size was 122.95 µm and span of 7.82. According to Nath et al., the size of filler is one of the most important factors that might affect the mechanical properties of the composite materials. Frequently, nanofiller with a size of 2 nm to 50 nm was used in synthesizing the composite hydrogel [32,33,34]. Large-sized filler might create a weak point in the material [35]. Several reports found to utilize raw FA in its pristine state without grinding and sieving [35,36].

#### 2.1.2. Fourier Transform Infrared Spectroscopy (FTIR)

Figure 2a shows the FTIR spectrum of FA. The broadband between 3718.99 to 2731.43 cm^−1^ is attributed to asymmetric and symmetric stretching vibration –OH, suggesting the presence of an amorphous silicate material or possibly hydrated aluminum silicates. The band at 1627.24 cm^−1^ is due to the bending mode of H_2_O molecules. A strong and broadband between 1548.96 to 823.20 cm^−1^ is due to Si–O–Si asymmetric stretching vibration. The absorbance bands at 789.41 and 473.97 cm^−1^ represent Si-O-Al bending vibration of octahedral Al and bending of Si–O, respectively. The spectra of raw FA is comparable to the previous studies [37,38].

Referring to Figure 2b, the additional FA in 0.5FA-AAc/AAm can be validated by observing the disappearance of C=C bending at 1100 and 936 cm^−1^. This can be used to presume that FA was successfully grafted into the AAc/AAm hydrogel. The IR spectra of 0.5FA-AAc/AAm+CO_2_ also displayed a few key differences at 1540 and 1320 cm^−1^ which correspond to the symmetric and asymmetric COO– stretching vibration. The presence of COO– stretching indicates the carbamate formation as an outcome from the reaction between MEA and CO_2_. Thus, this means that CO_2_ molecules could readily penetrate hydrogel and react with the infused MEA. Similar IR spectra with similar bands were observed in previous study by Xu et al. using DEA Infused Hydrogels (AIH) for CO_2_ capture [11]. Several common peaks were also observed between the AAc/AAm and 0.5FA-AAc/AAm+CO_2_ resulted from the use of copolymer of acrylic acid and acrylamide. Broad transmittance bands between 3600 cm^−1^ and 2750 cm^−1^ were assigned to -OH stretching which were originated by the hydroxyl group from carboxylic acid of acrylic acid. At 1650 cm^−1^, the common stretching of C=O groups were observed for both spectra. These align peaks were assigned to C=O group connected to the amide group of acrylamides.

#### 2.1.3. Scanning Electron Microscope (SEM)

The microstructure of AAc/AAm hydrogel and 0.5FA-AAc/AAm composite hydrogels are shown in Figure 3(a(i,ii),b(i,ii)). The morphology of both AAm/AAc and 0.5FA-AAc/AAm for Figure 3(ai,bi) were taken at 64× magnification while Figure 3(aii,bii) were taken at 5KX magnification. Based on the surface morphology of Figure 3(bi), it can be observed that there was no crack between FA and polymer matrix, which might signal the compatibility of the FA and AAc/AAm hydrogel. These morphologies were seen during the collapse state. As a result, the interlinkage matrices were obscured because of the shrinkage during the drying process and when the liquid is absorbed, the hydrogel can revive to its original form [39]. Anyhow, some pores still can be observed in Figure 3(aii,bii). With the addition of FA, as pointed by the arrows in Figure 3(bii), it is examined that the surface porosity volume of 0.5FA-AAc/AAm hydrogel was more than AAm/AAc hydrogel in Figure 3(aii). The surface of 0.5FA-AAc/AAm hydrogel exhibits tiny and denser pores in comparison to AAc/AAm hydrogel, which has wider diameter but only a few pores. FA presents a wide range of benefits due to its silica, alumina, and unburned carbon content which is useful as a precursor for the synthesis of porous materials.

#### 2.1.4. Thermogravimetric Analysis (TGA)

TGA was used to determine the effect of temperature on the desorption of CO_2_ and the evaporation of the MEA. The TGA analysis was carried out by heating up the samples from 25 to 600 °C. All samples were loaded with same amount of 30 wt% MEA and 0.5FA-AAc/AAm pre-saturated with CO_2_ to access the evaporation and desorption conditions. The TGA and DTG curves of AAc/AAm hydrogel, 0.5FA-AAc/AAm composite hydrogel and CO_2_ pre-saturated 0.5FA-AAc/AAm composite hydrogel are shown in Figure 4a–c. Combination evaporation of 30 wt% MEA which contain 70% of water was observed at the temperature <250 °C for all samples. From the TGA curves, more weight loss was recorded by 0.5FA-AAc/AAm composite hydrogel at the first stage of decomposition (temperature <250 °C) which shows that the incorporation of FA in the hydrogel can promote the absorption of amine. In Figure 4c, clear DTG peaks correspond to loss of water and MEA can be seen at 120 °C and 220 °C. While the desorption of CO_2_ occurred at 160 °C. A similar curve was also reported by Xu et al. [11]. The DTG curves show that the degradation of the polymer backbone or referred as hydrogel composite occurred at 380 to 420 °C, suggesting the hydrogels were thermally stable up until 380 °C. Base on the results, the hydrogel adsorbent can be regenerated until certain temperature and the water, MEA, and CO_2_ can be regenerated.

### 2.2. Carbon Dioxide Adsorption Kinetics

Figure 5 presents the kinetics data and the fitting plots for pseudo-first order, pseudo-second order, Weber-Morris, Alovich and Avrami fractional adsorption kinetic model and Table 2 tabulates the obtained kinetics form the models. The fitting calculation was carried out on the cumulative adsorption of the sample 30 wt% MEA infused 0.5FA-AAc/AAm which was performed at reaction temperature of 30 °C, CO_2_ gauge pressure of 2 bar and gas flow rate of 60 mL/min. The experimental and fitting results are shown in Figure 5.

It can be seen that from Table 2, the correlation coefficients, R^2^ of the pseudo-second order and Elovich kinetic model are 0.9905 and 0.9644, respectively. Meanwhile, the R^2^ of the Weber-Morris kinetic model is 0.8687. These kinetic models show relatively low R^2^ compared to pseudo-first order and Avrami kinetic model, so it is not fit to explain the kinetic behavior of 30 wt% MEA infused 0.5FA-AAc/AAm hydrogel. The Avrami kinetic model demonstrated a very close degree of correlation with the pseudo-first order kinetic model, as evidenced by R^2^ values of 0.9996 and 0.9999, respectively. However, on top of R^2^ analysis, other statistical indicators that represent residual error like SSE must be analyzed to validate the modelling [40]. Thus, upon analysis of the sum of squared estimate errors (SSE), it was determined that the pseudo-first order model was the most appropriate for describing the adsorption kinetics of CO_2_ by a 30 wt% MEA-infused 0.5FA-AAc/AAm hydrogel.

From this kinetic model, it can be said that the adsorption of CO_2_ is a reversible process. Assuming there was no dissociation of CO_2_ happened on the surface of 30 wt% MEA-infused 0.5FA-AAc/AAm hydrogel, the sorption phenomenon can be described as the diffusion-controlled process. Besides, if there are no adsorbate initially present on the adsorbent, the rate of change of CO_2_ uptake by the 30 wt% MEA-infused 0.5FA-AAc/AAm hydrogel with time is directly proportional to the difference in saturation concentration and the amount of CO_2_ uptake with time which is generally applicable over the initial stage of an adsorption process. The pseudo-first order was used to further investigate the adsorption kinetic under different conditions.

#### 2.2.1. Effect of FA Content

In order to study the influence of FA content on the CO_2_ adsorption kinetics, the adsorption reaction experiment was carried out under the condition of reaction temperature of 30 °C, CO_2_ flow rate of 60 mL/min and FA content of 0 to 2.0 wt%. The experimental and pseudo-first order fitting results are shown in Figure 6a–c. The 0FA-AAc/AAm (hydrogel without addition of FA) exhibited 0.46 mol/g adsorption capacity and 2.2 min breakthrough time. Incorporation of 0.5 wt% of FA into the hydrogel resulted in a 54% increase in adsorption capacity and breakthrough time, with values of 0.71 mol/g and 7 min, respectively. However, further increases in FA content led to a decline in adsorption capacity, with values of 0.57 mol/g and 0.39 mol/g for 1.0 and 2.0 wt% of FA, respectively. The kinetic constants for 0FA, 0.5FA, 1.0FA and 2.0FA-MEA infused hydrogel were determined at 1.54 min^−1^, 0.76 min^−1^, 1.01 min^−1^, and 1.41 min^−1^ respectively. The use of aluminosilicate bentonite in ethylcellulose in study by Alekseeva et al. (2020) also demonstrated a similar trend where the adsorption kinetic rate increased as the FA content increased, despite a decline in adsorption capacity [41].

FA is an aluminosilicate mineral, which means it consists primarily of silica, aluminum, iron and calcium oxide [42] and has the capability to adsorb CO_2_ [20]. In general, incorporating a nanocomposite or filler into a polymer adsorbent increases porosity compared to the neat polymer. But, in this instance, only 0.5 wt% concentration of FA improved the adsorption capacity in this situation. A higher FA concentration led to a reduction in CO_2_ capture. According to Zhu et al. (2019), once the hydrogel has been activated and swollen, the cavities and active sites produced in the polymer network by the solidified FA particle would increase the number of adsorption points [43]. The SiO_2_ aids in promoting the hydrogel to form a hydrogen bond with other monomers [35,44]. Excessive FA induces more crosslinked points, thus limiting the hydrogel’s porosity.

#### 2.2.2. Effect of AAc/AAm Hydrogel Activator

Hydrogel is a versatile material that exhibits unique properties such as shrinkage, swelling, and the ability to retain large amounts of liquids. The activation of the hydrogel, through the infusion of a liquid solution, causes the material to swell and expand. This swelling results in the creation of additional void spaces within the hydrogel matrix and imbuing it with additional properties based on the properties of infused liquid [43]. The swelling properties of the hydrogel are highly affected by the ionic strength, pH and temperature. It is important to investigate the most effective hydrogel activator in order to maximize the adsorption of CO_2_. 0.5FA-AAc/AAm hydrogel exhibited 12% swelling percentage in 100 wt% MEA. Contrarily, a significant amount of water (800%) and 30 wt% MEA solution (1150%) were able to be infused in the 0.5FA-AAc/AAm hydrogel matrices as presented in Figure 7. This implies that a smaller amount of hydrogel is required to support the same amount of pure and 30 wt% amine solution and the utilization of dry hydrogel also can be minimized during adsorption process. The water and MEA solution, which had been infused into the hydrogel matrix, served as the activating agent for the CO_2_ capture process, effectively adsorbing CO_2_ from the surrounding environment.

As shown in Figure 8a,b, at temperature of 30 °C, pressure of 2 bar and flowrate of 60 mL/min, the CO_2_ adsorption capacity of water infused 0.5FA-AAc/AAm hydrogel was 0.51 mol/g. A further analysis of the kinetic model revealed that the pseudo-first order kinetic rate increases in the following order: 30 wt% MEA < water < 100 wt% MEA, with values of 0.76 min^−1^, 0.96 min^−1^, and 0.98 min^−1^, respectively, as demonstrated in Figure 8c and summarized in Table 3.

Water exhibited a comparable adsorption capacity to the 100 wt% MEA infused hydrogel. 1 mol of diffused CO_2_ can react with 1 mol of hydroxyl groups to form carbonic acid (shown in Equation (1)), and 1 mol of amine (C_2_H_7_NO) can only react with 0.5 mol of CO_2_ (shown in Equation (2)), but the nitrogen atom in amine molecules is more electronegative and attractive towards CO_2_. Despite the high adsorption of water creating more void within the hydrogel adsorbent, the use of amine as the activator was found to be more effective in adsorbing CO_2_, supported by the slightly higher kinetic rate of the 100 wt% MEA-infused hydrogel. Thus, it is reasonable to assume that pure amine infused hydrogel would be more efficient in capturing CO_2_ than one infused with water.

Meanwhile, the adsorption capacity shows a significant increment with the use of 30 wt% of MEA solution. It also can be inferred that an increase in the concentration of MEA leads to an increase in the kinetic rate, indicating a faster reaction with CO_2_. However, with the presence of water in amine solution, 1 mol of MEA now can react with 1 mol of CO_2_ and form bicarbonate, as shown in Equation (3), resulting higher CO_2_ adsorption capacity.
CO_2_ + H_2_O ↔ HCO_3‾_ + H^+^(1)
CO_2_ + 2 HOC_2_H_4_NH_2_ ↔ HOC_2_H_4_NHCO_2‾_ + HOC_2_H_4_NH_3_^+^(2)
CO_2_ + HOC_2_H_4_NH_2_ + H_2_O ↔ HCO_3‾_ + HOC_2_H_4_NH_3_^+^(3)

#### 2.2.3. Effect of Operational Temperature

A series of adsorption experiments were investigated in fixed bed column at 30, 45 and 60 °C. Figure 9 shows the breakthrough curve and adsorption capacity of 0.5FA-AAc/AAm at three different operational temperature. It shows that the higher adsorption temperature, the shorter the break curve and saturation time. At adsorption temperature of 30 °C, the CO_2_ adsorption amount of 0.5FA-AAc/AAm was 0.71 mol/g. However, it decreased to 0.53 and 0.49 mol/g at 45 and 60 °C, respectively. From the Table 3, it can see that the kinetic constant, k_1_ increases from 0.76 min^−1^ to 1.14 min^−1^ as the adsorption temperature increase from 30 °C to 60 °C, indicating that the increase in temperature promotes the reaction rate. In general, as temperature increases, the kinetic energy of gas particles also increases, leading to more frequent collisions between gas and adsorbent molecules and resulting in a higher adsorption capacity. Conversely, the reaction rate and CO_2_ equilibrium adsorption capacity were inversely proportional to each other. At temperatures above 60 °C, thermodynamic effect become more significant in controlling the reaction compared to kinetic effect, and the carbamate molecules begin to reverse back to CO_2_ as shows in Equation (14), resulted the decrease in equilibrium adsorption capacity [5,25].

The CO_2_ adsorption of FA-AAc/AAm hydrogel in this study varied from 0.39 to 0.71 mol/g depending on the FA content, type of hydrogel activator and operational temperature. It is noted that the use of different amine, starting monomer, filler or polymerization technique will give different properties to the adsorbent. For instance, MEA-hydrogel with the dodecyltrimethylammonium bromide (DTAB) surfactant showed an adsorption of 2.96 mmol/g [8]. Meanwhile, silica that functionalize with 3-aminopropyltriethoxysilane (APTES) via wet grafting method able to capture CO_2_ up to 1.67 mmol/g [45]. Another example of amine infused adsorbent is APTES functionalized aluminosilicate; which came from metakaolin and silica fume. The study successfully absorbed 1.17 mmol/g of CO_2_ and the adsorbent also exhibited low adsorption and desorption energy of 14.47 and 51.05 kJ/mol [46]. The concept of incorporating amine in solid adsorbent has been established for some time, but the search for the best adsorbent in term of sorption capacity, industrial pliability and sustainable for a long run should be continue with more material to be explored.

## 3. Conclusions

The FA-AAc/AAm hydrogel was synthesized via in-situ solution polymerization and hydrogel activator was infused by direct impregnation method in which the hydrogel was immersed in the water or MEA solution for 24 h. The morphology of the hydrogel featured less visible pores but demonstrated a high capacity for hydrogel activator absorption. It is suggested that the amine absorption properties of the hydrogel are also influenced by the functional groups present. The incorporation of fly ash into AIHs was found to effectively improve CO_2_ adsorption. At a temperature of 30 °C and pressure of 2 bar, the use of 0.5 wt% of FA increased CO_2_ adsorption by 54% compared to neat AIHs. Data from this study were best simulated by the pseudo-first order > Avrami fractional > pseudo-second order > Elovich > Weber-Morris kinetic models, as supported by the SSE value. It can be inferred that the rate of change in CO_2_ uptake with respect to time is directly proportional to the difference in saturation concentration and the amount of solid uptake. However, the Avrami fractional kinetic model cannot be completely disregarded due to only 0.99% difference in SSE between both kinetic models. Although the hydrogel material is stable at high temperature up to 380 °C, the breakthroughs analysis suggest that the adsorption of CO_2_ is not favorable at high temperatures. As a result, to optimize the adsorption capacity of the amine infused FA-AAc/AAm hydrogel, it is recommended to lower the flue gas temperature prior to the adsorption. Further optimization studies, such as varying the FA composition, FA particle size, type of amine solution, operational pressure, and increasing temperature, are also recommended in order to maximize CO_2_ capture. With further exploration of these remaining factors, this easily synthesized AIHs has the potential to become a commercially viable adsorbent for CO_2_ capture.

## 4. Material and Methods

### 4.1. Materials

Acrylamide (analytical reagent grade) and acrylic acid (99% purity) were supplied by Merck. Ammonium peroxodisulphate (analytical reagent grade) was supplied by R&M Chemicals. *N*,*N*’-methylenebis(acrylamide) (99% purity), was purchased from Sigma-Aldrich. A synthesis-grade of monomers and cross-linker were used as-received without further purification. Distilled water was used throughout this research work as the polymerization medium. FA was obtained from a coal power plant located in Northern Malaysia.

### 4.2. Preparation of MEA Infused FA-AAc/AAm Hydrogel Adsorbent

The amine infused hydrogel was synthesized by solution polymerization using common monomers, acrylic acid (AAc) and acrylamide (AAm). The schematic illustration of synthesis process and CO_2_ adsorption is shown in Figure 10. The synthesis process started by diluting 5 g of AAm into 100 mL distilled water. The solution was then combined with sieved FA ranging from 0.5 wt% to 2 wt%. The stirring continued for another 20 min with temperature maintained at 60 °C. Then, 0.5 g of AAc and 0.05g of *N*,*N*”-methylene bisacrylamide (NMBA) as cross-linking agent were introduced to the solution while the mixture was continuously stirred. The initiator, 0.25 g of ammonium peroxodisulphate (APS) was added after 30 min. The FA-AAc/AAm hydrogel was dried at 55 °C for 8 h in the vacuum oven before it was infused with activator namely water, 100 wt% monoethanolamine (MEA) and 30 wt% MEA solution.

### 4.3. Characterization

The elemental composition of raw FA was analyzed by energy dispersive XRF (X-ray fluorescence spectrometry, Shimadzu XRF-1700, Tokyo, Japan) according to BS EN ISO 12677:2011 method. The sampling of FA was conducted by taking the FA from five different sampling points. The FA was ground using a planetary ball mill to obtain the particle size of less than 100 µm prior to mounting into the sample holder.

Malvern Instrument Mastersizer 3000 was used to determine the particle size distribution (PSD) of FA before and after the ball-mill grinding. FA samples were analyzed in triplicates and an average result recorded. The particle refractive index and adsorption index was set at 1.6 and 1.0, respectively.

The functional groups of raw FA and the effect of incorporation of FA on the hydrogel matrices was determined using FTIR spectra model Perkin-Elmer Spectrum One, FTIR-frontier. The FTIR spectra was determined at wavenumber ranges, scan number per sample and wave resolution of 4000–500 cm^−1^, 16 and 4 cm^−1^, respectively. The hydrogel samples were prepared using vacuum oven-dried for 5 h at 50 °C. The pellets for FTIR analysis were prepared from a uniform mixture of hydrogel and potassium bromide (KBR) using a steel die. KBr was used as the sample carrier because its optically transparent properties would not impose any absorbance interference to the IR ray.

The microstructure of hydrogel samples was determined using Scanning electron microscopy (SEM, Zeiss Evo LS15, Munich, Germany) with an energy of 10 keV at various magnification from ×60 to ×5000. The hydrogels samples were dried using a freeze dryer for 48 h to preserve the hydrogel structure. The samples were coated with a thin layer of gold to add the conductivity into the insulted hydrogel surface before analysis.

The AAc/AAm, 0.5FA-AAc/AAm and CO_2_ loaded 0.5FA-AAc/AAm samples were put into the sample pan of a Perkin Ekmer STA6000 thermal gravimetric analyzer. Waltham, MA USA The chamber was then purged with nitrogen gas before it was heated from 20 to 600 °C at a heating rate of 10 °C/min. The system temperature, sample mass and mass derivative rate as a function of time were recorded throughout the heating process.

### 4.4. CO_2_ Adsorption Analysis

To study the influencing factors of FA-AAc/AAm hydrogel on the CO_2_ adsorption performance, the experiments were conducted using 10 mm inner diameter and 100 mm high adsorption column. The setup of the self-assembled CO_2_ adsorption column is shown in Figure 11. Firstly, 1 g of MEA infused FA-AAc/AAm was packed into the stainless-steel column supported with glass wool on both ends. Then, the operational pressure was regulated to 2 bar and the temperature set to 30, 45 and 60 °C, depending on the predetermined experimental parameter. After that, the CO_2_ was passed into the adsorption column at a flow rate of 60 mL/min and the outlet CO_2_ concentration was automatically analyzed and recorded by the computer. The adsorption capacity of CO_2_ was calculated by the follow equation: (4)$$Q=\frac{{\mathrm{Ft}}_{\mathrm{ca}}C_{\mathrm{in}}}{m};\;t_{\mathrm{ca}}=\int_{0}^{t}\left(1-\frac{C_{\mathrm{eff}}}{C_{\mathrm{in}}}\right)\mathrm{dt}$$
where, Q is the adsorption capacity (mol/g), F is inlet molar flow rate (mol/min), C_in_ and C_eff_ represent the inlet and outlet concentration and m stands for the mass of adsorbent (g).

### 4.5. Adsorption Kinetic

The adsorption kinetic models were used to analyze the performance of the adsorbent and to predict the adsorption mass transfer mechanism. Several models are available to describe the adsorption of adsorbate on the adsorbent. Table 4 lists five these models and their corresponding linear expressions. One commonly used model is the pseudo-first order kinetic model, which is often employed to study the gas-solid adsorption processes that are controlled by surface diffusion [25]. The pseudo-second order kinetic model assumes that the rate of adsorption is limited by chemical sorption or chemisorption. The k_1_ in Equations (5) and (6) represents the rate constant, meanwhile k_2_ in Equations (7) and (8) denotes the pseudo-second order rate constant.

The Weber-Morris kinetic model or also known as intra-particle diffusion model suggest that the rate of adsorption is governed by the availability of active sites on the adsorbent surface rather than concentration of adsorbate. Referring to Equations (9) and (10), k_id_ represents the kinetic rate constant. The Elovich kinetic model was initially applied in gaseous systems to predict the mass and surface diffusion, activation and deactivation energy of the system. Referring to Equations (11) and (12), α is initial adsorption rate and β is desorption constant. The Avrami fractional kinetic model is a fractional kinetic with order one for particle nucleation and applicable for describing the adsorption of multicomponent gases by solid adsorbent [25]. In Equations (13) and (14), k_a_ is the Avrami rate constant, and n is the Avrami exponent which use to reflect the dimensionality of crystal growth [47]. The CO_2_ adsorption capacity data of 30 wt% MEA infused 0.5FA-AAc/AAm hydrogel was used to fit into all these models and determine the most suitable kinetic model to explain the kinetic behavior of CO_2_ adsorption of MEA infused hydrogel adsorbent.

## Figures and Tables

**Figure 1 gels-09-00229-f001:**
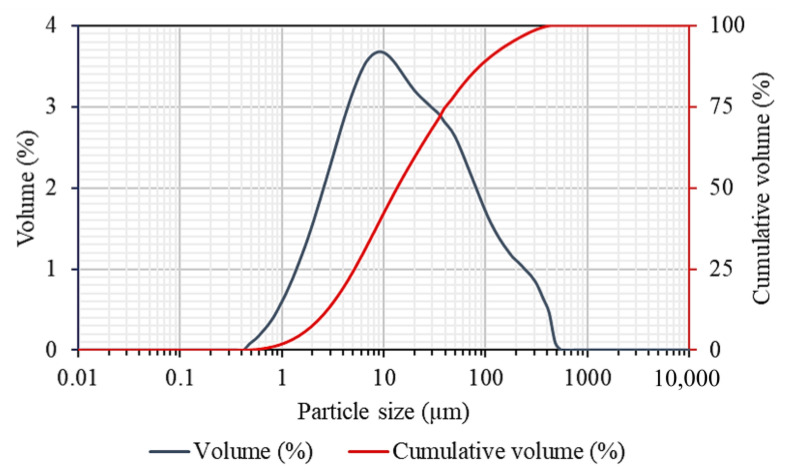
Particle size distribution of FA.

**Figure 2 gels-09-00229-f002:**
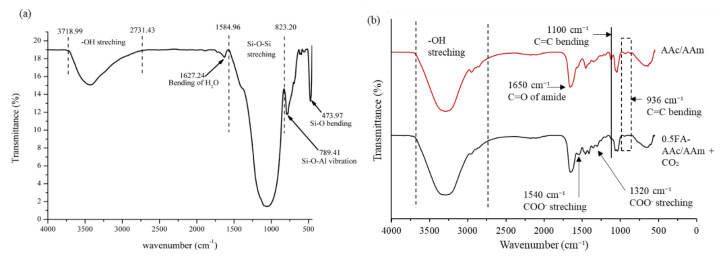
The FTIR spectra (**a**) FA and (**b**) the AAc/AAm hydrogel and 0.5FA-AAc/AAm+CO_2_ (hydrogel after CO_2_ capture).

**Figure 3 gels-09-00229-f003:**
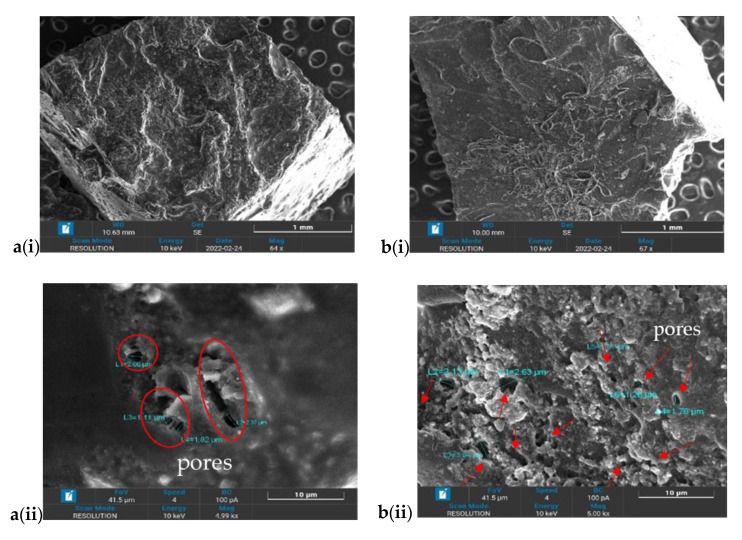
SEM images of (**a**) AAc/AAm and (**b**) 0.5FA-AAc/AAm at (**i**) 64× magnification and (**ii**) 5000× magnification; the red circles and arrows points the pores on the hydrogel surface.

**Figure 4 gels-09-00229-f004:**
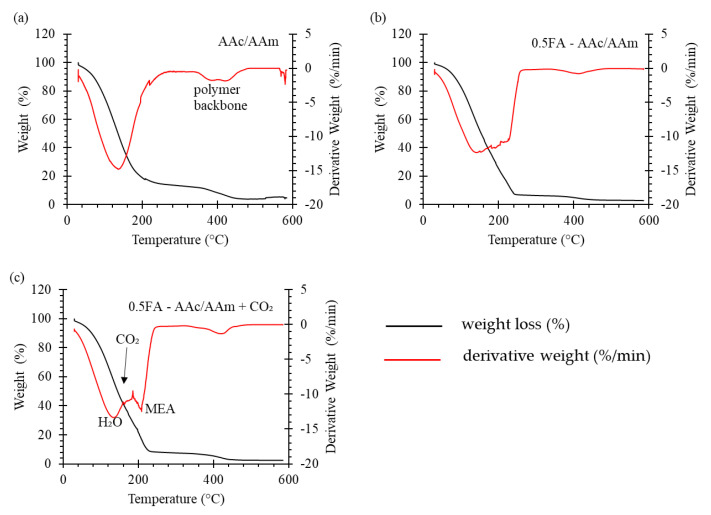
TGA and DTG curves of (**a**) AAc/AAm hydrogel, (**b**) 0.5FA-AAc/AAm composite hydrogel and (**c**) CO_2_ pre-saturated 0.5FA-AAc/AAm composite hydrogel.

**Figure 5 gels-09-00229-f005:**
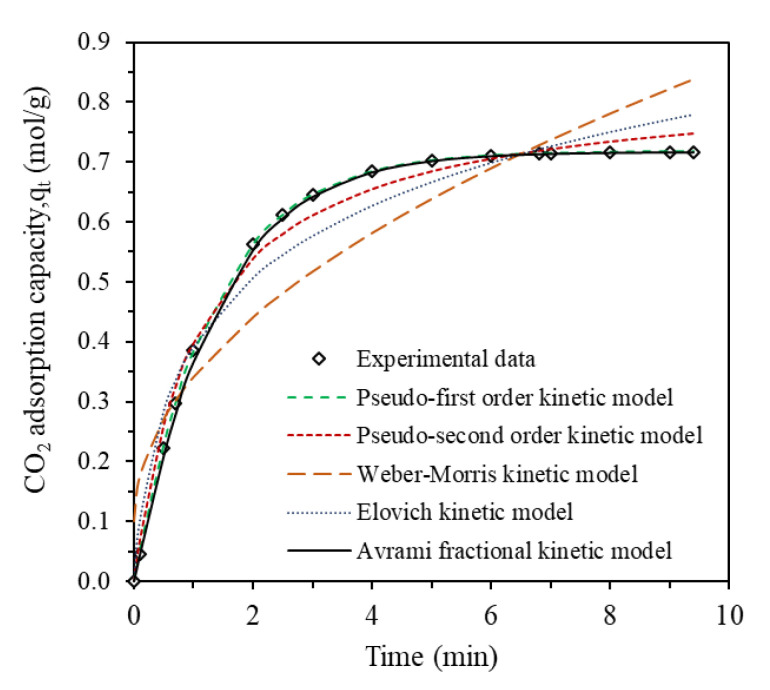
Fitting results of different adsorption kinetic model.

**Figure 6 gels-09-00229-f006:**
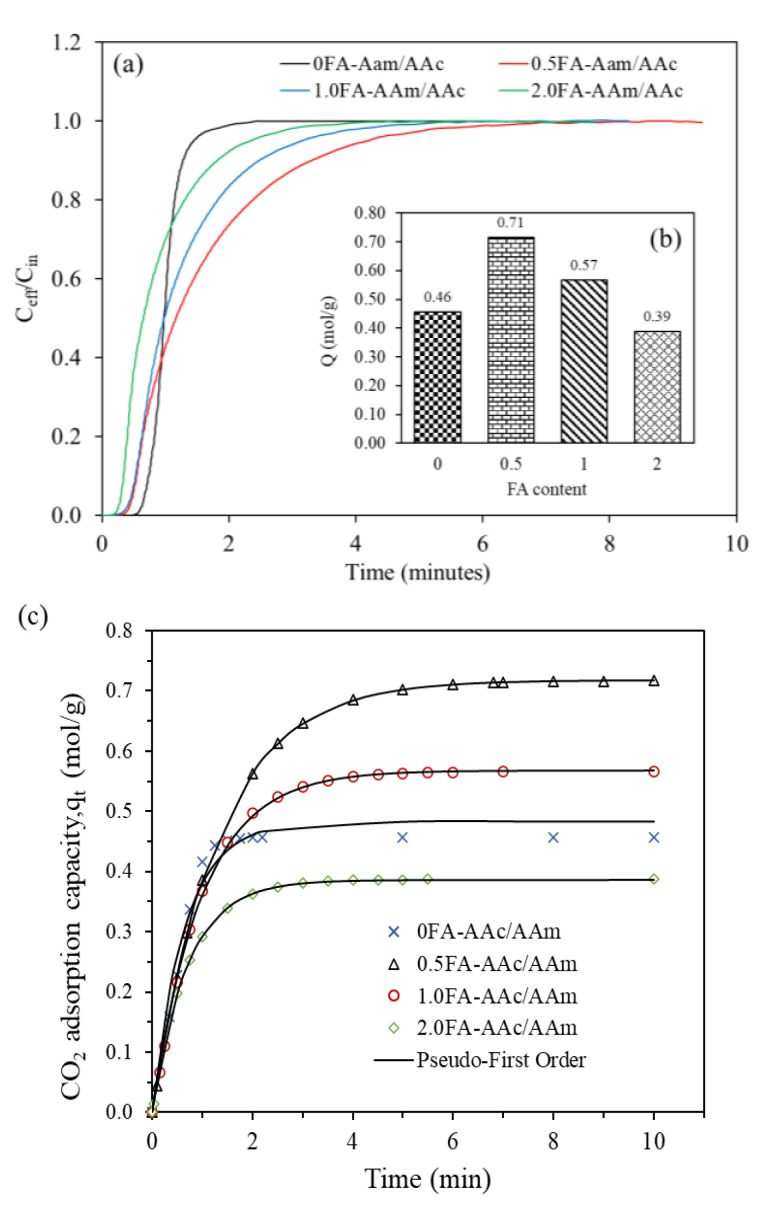
(**a**) Breakthrough adsorption curves, (**b**) equilibrium adsorption capacity and (**c**) pseudo-first order fitted adsorption capacity at different FA content (temperature = 30 °C, pressure = 2 bar, CO_2_ flowrate = 60 mL/min).

**Figure 7 gels-09-00229-f007:**
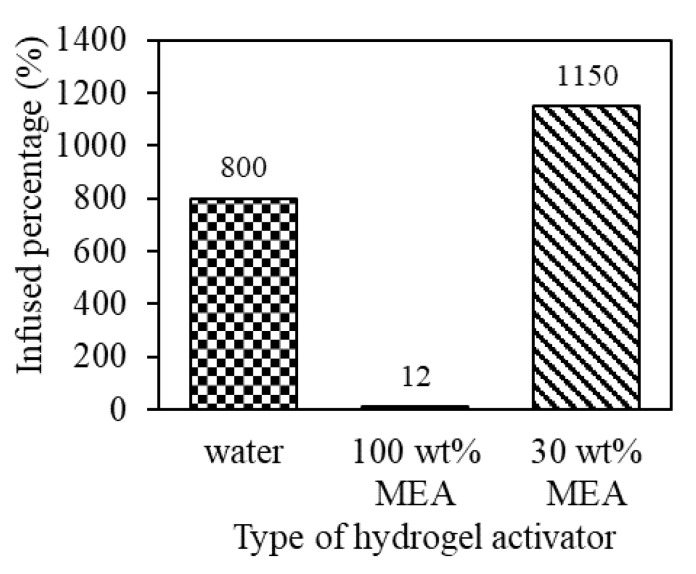
Water, 100 wt% MEA and 30 wt% MEA infused percentage in 0.5FA-AAc/AAm hydrogel.

**Figure 8 gels-09-00229-f008:**
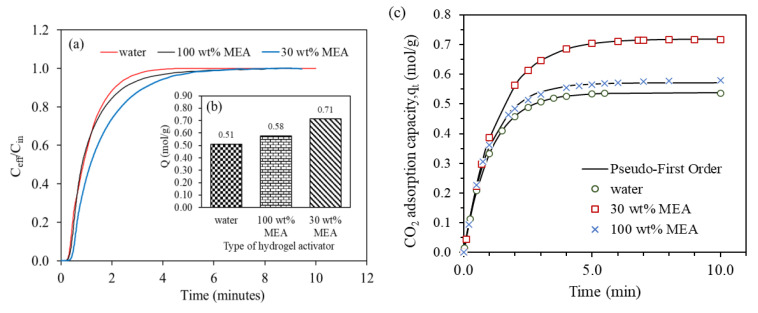
(**a**) Breakthrough adsorption curves, (**b**) equilibrium adsorption capacity and (**c**) pseudo-first order fitted adsorption capacity in the presence of different hydrogel activators (temperature = 30 °C, pressure = 2 bar, CO_2_ flowrate = 60 mL/min).

**Figure 9 gels-09-00229-f009:**
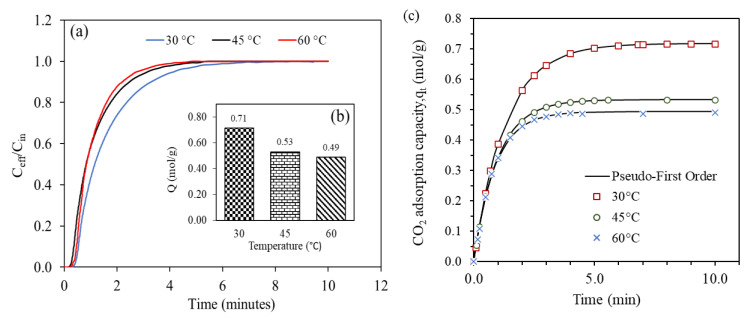
(**a**) Breakthrough adsorption curves, (**b**) equilibrium adsorption capacity and (**c**) pseudo-first order fitted adsorption capacity at different adsorption temperature (pressure = 2 bar, CO_2_ flowrate = 60 mL/min).

**Figure 10 gels-09-00229-f010:**
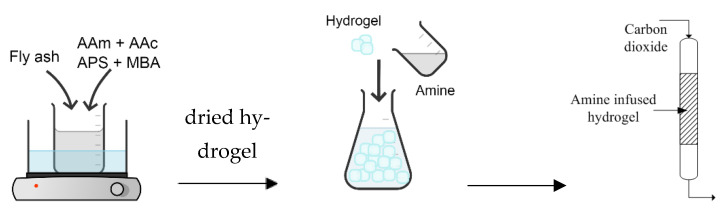
Illustration of synthesis and CO_2_ adsorption of MEA infused FA-AAc/AAm hydrogel adsorbent.

**Figure 11 gels-09-00229-f011:**
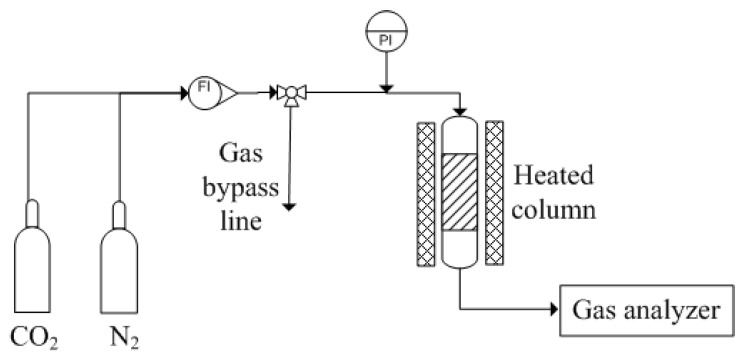
Schematic diagram of experiment setup.

**Table 1 gels-09-00229-t001:** Chemical composition of raw coal power plant FA.

Component	SiO_2_	Al_2_O_3_	Fe_2_O_3_	CaO	MgO	C	K_2_O	SO_3_	Others
Composition (%)	44.52	22.48	11.58	9.84	3.92	2.54	1.73	1.35	2.05

**Table 2 gels-09-00229-t002:** Fitting parameters of different adsorption kinetic model.

	Pseudo-First Order	Pseudo-Second Order	Weber-Morris	Elovich	Avrami
SSE (%)	0.01	0.92	12.68	3.44	1.00
R^2^	0.9999	0.9905	0.8687	0.9644	0.9996
q_e_ (mol/g)	0.7176	0.7475	0.8381	0.7800	0.7164
Constant	k_1_ = 0.7637 min^−1^	k_2_ = 1.0872 mol g^−1^ min^−1^	k_id_ = 0.2403 mol (g min^1/2^)	α = 1.3830 mol g^−1^ min^−1^ β = 5.48941 g mol^−1^	0.7587 min^−1^
n	-	-	-	-	1.0476

**Table 3 gels-09-00229-t003:** Summarize of pseudo-first order rate constant, k_1_ at different CO_2_ adsorption condition.

Condition	Parameter	k_1_ (min^−1^)
FA content	0	1.54
	0.5	0.76
	1.0	1.01
	2.0	1.41
Hydrogel activator	water	0.96
	30 wt% MEA	0.76
	100 wt% MEA	0.98
Operational temperature	30 °C	0.76
	45 °C	1.01
	60 °C	1.14

**Table 4 gels-09-00229-t004:** Adsorption kinetic models in linear and non-linear expressions.

Models	Equation		Linear Expression	
Pseudo-first order kinetic model	$q_{t}=q_{e}(1-\;e^{-{\mathrm{tk}}_{1}})$	(5)	ln(q_e_ − q_t_) = ln(q_e_) − k_1_t	(6)
Pseudo-second order kinetic model	$q_{t}=\frac{q_{e}^{2}k_{2}t}{{1+q}_{e}k_{2}t}$	(7)	$\frac{t}{q_{t}}=\frac{1}{k_{2}q_{e}^{2}}+\frac{t}{q_{e}}$	(8)
Weber-Morris kinetic model	q_t_ = k_id_t^1/2^ + C	(9)	q_t_ = k_id_t^1/2^ + C	(10)
Elovich kinetic model	$q_{t}=\frac{1}{\beta }\ln\;(1+\alpha \beta t)$	(11)	$q_{t}=\frac{1}{\beta }\ln(\alpha \beta )$ + $\frac{1}{\beta }\ln(t)$	(12)
Avrami fractional kinetic model	$q_{t}=q_{e}\left(1-e^{-\frac{{\left(k_{a}t\right)}^{n}}{n}}\right)$	(13)	$\ln\left[\ln\left(\frac{q_{e}}{q_{e}-q_{t}}\right)\right]$ = nln(k_a_) + nln(t)	(14)

q_t_ is adsorption capacity at time t; q_e_ = equilibrium adsorption capacity; t is time.

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
