# Peer review of "Amine Infused Fly Ash Grafted Acrylic Acid/Acrylamide Hydrogel for Carbon Dioxide (CO2) Adsorption and Its Kinetic Analysis"

_gels, 2023, doi:10.3390/gels9030229_

Round 1

Reviewer 1 Report

In this work, newly explore FA grafted AIHs was fabricated via direct impregnation of MEA in FA grafted AAc/AAm hydrogel. The impact of FA content, type of hydrogel activator, and operational temperature on CO2 capture was systematically evaluated in order to maximize the CO2 adsorption capacity.

This is an interesting work; Nevertheless some revisions are needed in order to publish this manuscript.

1. First of all, there are some errors in the reference list of the paper (Reference source not found..) Please correct.

2. The authors state that the currently used FA was classified as a
Class F FA as described in ASTM C618. This means it
contains more 70 % of Al2O3, SiO2 and ferrite and only a small amount of other chemical compositions including calcium (Ca).

The authors need to analyze more how they calculated the contents of their FA. FT-IR is not enough. could they compare more their samples with the literature?

3. Could the authors perform porosimetry measurements? I feel this information is crusial. Could they comment on the BET of their samples?

4. could the authors comprare their findings with other manuscripts working on FA?

Author Response

Response to Reviewer 1 Comments

Generel remarks from reviewer. In this work, newly explore FA grafted AIHs was fabricated via direct impregnation of MEA in FA grafted AAc/AAm hydrogel. The impact of FA content, type of hydrogel activator, and operational temperature on CO2 capture was systematically evaluated in order to maximize the CO2 adsorption capacity.

This is an interesting work; Nevertheless some revisions are needed in order to publish this manuscript.

Point 1. First of all, there are some errors in the reference list of the paper (Reference source not found..) Please correct.

Response 1. The errors were fixed accordingly.

Point 2. The authors state that the currently used FA was classified as a
Class F FA as described in ASTM C618. This means it contains more 70 % of Al2O3, SiO2 and ferrite and only a small amount of other chemical compositions including calcium (Ca).

The authors need to analyze more how they calculated the contents of their FA. FT-IR is not enough. could they compare more their samples with the literature?

Response 2. The composition of FA was analyse using X-ray Fluorescence Spectrometry (XRF) and all the composition presented was computer generated value. The author also added comparison with other literature as stated “While some analysis reported that the composition of SiO2 and Al2O3 can be up to 55% and 35% respectively [29,30] ”.

Point 3. Could the authors perform porosimetry measurements? I feel this information is crucial. Could they comment on the BET of their samples?

Response 3. Yes, as mentioned by the reviewer, the porosity of the hydrogel is one of the curial info to be discussed when it is related to the adsorption. However, because of the nature of the synthesized hydrogel, the authors did not discuss the BET results. The hydrogel's surface structure, as seen in Figure 3, is very dense with only a small number of pores present during its dry/shrink state. Despite several BET analyses were conducted, the results only indicated a very low porosity caused by shrinkage after drying. The CO2 adsorption was performed on the hydrogel in its wet state, thus, the authors also believe that the BET analysis results on its dry state may not accurately represent the hydrogel's porosity and surface area during the adsorption. Although some researchers have reported their findings on BET of hydrogels, this study's hydrogel exhibited different behavior from the previous studies in terms of BET results. Therefore, instead of using BET analysis, the authors prefer to use thermoporometry analysis to determine the hydrogel's porosity in its wet state. Unfortunately, the currently available facility does not allow for thermoporometry to be conducted.

Point 4. could the authors compare their findings with other manuscripts working on FA?

Response 4. The author could not find the similar study in term of the use of FA-hydrogel as adsorbent to be used as comparison even though literature search with various keyword has been done. Thus, the author compared the finding with other amine infused solid support. A paragraph for the finding comparison was added at the end of results and discussion; as follow:

“The CO2 adsorption of FA-AAc/AAm hydrogel in this study varied from 0.39 to 0.71 mol/g depending on the FA content, type of hydrogel activator and operational temperature. It is noted that the use of different amine, starting monomer, filler or polymerization technique will give different properties to the adsorbent. For instance, MEA-hydrogel with the dodecyltrimethylammonium bromide (DTAB) surfactant showed an adsorption of 2.96 mmol/g [8]. Meanwhile, silica that functionalize with 3-aminopropyltriethoxysilane (APTES) via wet grafting method able to capture CO2 up to 1.67 mmol/g [47]. Another example on amine infused adsorbent is APTES function-alized aluminosilicate; which came from metakaolin and silica fume. The study suc-cessfully adsorbed 1.17 mmol/g of CO2 and the adsorbent also exhibited low adsorption and desorption energy of 14.47 and 51.05kJ/mol. The concept of incorporating amine in solid adsorbent has been established for some time, but the search for the best adsor-bent in term of sorption capacity, industrial pliability and sustainable for a long run should be continue with more material to be explored.”

Reviewer 2 Report

The paper proposed is interesting, carried out carefully and with the appropriate scientific investigations. 

I have just a few changes to suggest:

- Line 32, in the cited document, coal-fired plants appear to be about half, check it

- "Error! Reference source not found" often appears

- In Table 1 it would be better to write "chemical composition" rather than "elemental"

- Looking at the TGA curves and the breakthrough curves, the question arises: How would the material behave in real plant conditions? Should emissions be cooled? and for an ideal plant of such as 500kw how efficiently would adsorb the CO2 gel and for how long. It would be useful to include these estimates in the conclusions to make the work more complete.

- Paragraph "4.4 CO2 adsorption analysis". This paragraph contains the formula used to calculate adsorption. in the calculation of the piers has been taken into account that the system operates at a different pressure than the environmental one (which is that of input and output of CO2)? check the calculations

Author Response

Response to Reviewer 2 Comments

General remarks from reviewer. The paper proposed is interesting, carried out carefully and with the appropriate scientific investigations. I have just a few changes to suggest:

Point 1. Line 32, in the cited document, coal-fired plants appear to be about half, check it

Response 1. The miscalculated “4492” power plant was rechecked and changed to correct the value “2215”.

Point 2. "Error! Reference source not found" often appears

Response 2. All the errors were corrected.

Point 3. In Table 1 it would be better to write "chemical composition" rather than "elemental"

Response 3. After considering the feedback form other reviewer too, the “chemical composition” was change to “composition (%)”

Point 4. Looking at the TGA curves and the breakthrough curves, the question arises: How would the material behave in real plant conditions? Should emissions be cooled? and for an ideal plant of such as 500kw how efficiently would adsorb the CO2 gel and for how long. It would be useful to include these estimates in the conclusions to make the work more complete.

Response 4. As per reviewer suggestion, “Although the hydrogel material is stable at high temperature up to 380 °C, the breakthroughs analysis suggest that the adsorption of CO2 is not favorable at high temperature. As a result, to optimize the adsorption capacity of the amine infused FA-AAc/AAm hydrogel, it is recommended to lower the flue gas temperature prior to the adsorption.” Was added into the conclusion to give more insight to the reader.

Point 5.  Paragraph "4.4 CO2 adsorption analysis". This paragraph contains the formula used to calculate adsorption. in the calculation of the piers has been taken into account that the system operates at a different pressure than the environmental one (which is that of input and output of CO2)? check the calculations

Response 5. This experiment was run at constant pressure, 2 bar instead of atmosphere pressure due to equipment constrain. Based on the equation, the pressure of reaction should not take into account during the calculation. The formula used was also confirmed with other publication, for instance “Liu, F.; Fu, W.; Chen, S. Synthesis, characterization and CO2 adsorption performance of a thermosensitive solid amine adsorbent. Journal of CO2 Utilization 2019; 31:98–105. https://doi.org/10.1016/j.jcou.2019.02.019.”

Reviewer 3 Report

The present study describes the adsorption of carbon dioxide using a novel amine-infused hydrogel material. The investigation of CO2 adsorption in this material is of significant interest to researchers focused on amine-based CO2 absorption, as it presents a new avenue for achieving enhanced performance. To better understand the adsorption behavior, the newly developed material was characterized using several techniques, including Fourier transform infrared (FTIR) spectroscopy, scanning electron microscopy (SEM), and thermogravimetric analysis (TGA). The adsorption kinetics of CO2 in the amine-infused hydrogel were also carefully investigated using several mathematical models. The results of these investigations showed that the adsorption of CO2 was efficient and effective. The manuscript is well-written overall and provides valuable insights into the use of the novel amine-infused hydrogel for CO2 adsorption. Therefore, I would recommend the manuscript for publication pending the authors address some minor concerns raised below.

1. The elemental composition of the sample is presented in Table 1, but the results are given in terms of oxide formulas. In addition to this, it would be beneficial if the authors could also provide the elemental composition in terms of the actual elements present in the sample.

 2. There are several typos and referencing errors throughout the manuscript that should be addressed. For example, there are two Table 1s in the manuscript, which could cause confusion for readers. Careful proofreading and editing can help to correct these errors and improve the overall clarity of the manuscript.

 3. Another Table 1 presents the fitting parameters for different adsorption models, but there seems to be a discrepancy with the experimental data presented in Figure 5. Some of the models appear to be significantly off the experimental data, even though Table 1 indicates a good fit with a SSE and a high R2. It would be beneficial for the authors to double-check the data and ensure that the presented results are accurate and consistent across all sections of the manuscript

Author Response

Response to Reviewer 3 Comments

General remarks from reviewer. The present study describes the adsorption of carbon dioxide using a novel amine-infused hydrogel material. The investigation of CO2 adsorption in this material is of significant interest to researchers focused on amine-based CO2 absorption, as it presents a new avenue for achieving enhanced performance. To better understand the adsorption behavior, the newly developed material was characterized using several techniques, including Fourier transform infrared (FTIR) spectroscopy, scanning electron microscopy (SEM), and thermogravimetric analysis (TGA). The adsorption kinetics of CO2 in the amine-infused hydrogel were also carefully investigated using several mathematical models. The results of these investigations showed that the adsorption of CO2 was efficient and effective. The manuscript is well-written overall and provides valuable insights into the use of the novel amine-infused hydrogel for CO2 adsorption. Therefore, I would recommend the manuscript for publication pending the authors address some minor concerns raised below.

Point 1. The elemental composition of the sample is presented in Table 1, but the results are given in terms of oxide formulas. In addition to this, it would be beneficial if the authors could also provide the elemental composition in terms of the actual elements present in the sample.

Response 1: The “elemental composition” was changed to “composition (%)” to avoid confusion for the readers

 Point 2. There are several typos and referencing errors throughout the manuscript that should be addressed. For example, there are two Table 1s in the manuscript, which could cause confusion for readers. Careful proofreading and editing can help to correct these errors and improve the overall clarity of the manuscript.

Response 2: The table numbers and figures were fixed accordingly and the cross-referencing were updated according to respective tables and figures.

Point 3. Another Table 1 presents the fitting parameters for different adsorption models, but there seems to be a discrepancy with the experimental data presented in Figure 5. Some of the models appear to be significantly off the experimental data, even though Table 1 indicates a good fit with a SSE and a high R2. It would be beneficial for the authors to double-check the data and ensure that the presented results are accurate and consistent across all sections of the manuscript

Response 3: Based on the comment, the author believes the concern model is Weber-Morris model. As reviewer suggested, the author has double-checked the data and ensured that the presented results are accurate and consistent across all section of the manuscript according to the fitting method used. The author utilised origin software to fit the modelling data and upon re-checking, similar fitting results was generated.